# Ethnic inequalities in COVID-19 vaccine uptake and comparison to seasonal influenza vaccine uptake in Greater Manchester, UK: A cohort study

Ruth Elizabeth Watkinson[1]*, Richard Williams[2], Stephanie Gillibrand[1], Caroline Sanders[3], Matt Sutton[1]

1 Health Organisation, Policy and Economics, Centre for Primary Care and Health Services Research, University of Manchester, Manchester, United Kingdom, 2 NIHR Greater Manchester Patient Safety Translational Research Centre, Manchester Academic Health Science Centre, University of Manchester, Manchestehr, United Kingdom, 3 NIHR Applied Research Collaboration Greater Manchester, Centre for Primary Care and Health Services Research, University of Manchester, Manchester, United Kingdom

* ruth.watkinson@manchester.ac.uk

**Data Availability Statement:** The legal basis for use of patient data in this study was defined in the

## Abstract

### Background

COVID-19 vaccine uptake is lower amongst most minority ethnic groups compared to the White British group in England, despite higher COVID-19 mortality rates. Here, we add to existing evidence by estimating inequalities for 16 minority ethnic groups, examining ethnic inequalities within population subgroups, and comparing the magnitudes of ethnic inequalities in COVID-19 vaccine uptake to those for routine seasonal influenza vaccine uptake.

### Methods and findings

We conducted a retrospective cohort study using the Greater Manchester Care Record, which contains de-identified electronic health record data for the population of Greater Manchester, England. We used Cox proportional hazards models to estimate ethnic inequalities in time to COVID-19 vaccination amongst people eligible for vaccination on health or age (50+ years) criteria between 1 December 2020 and 18 April 2021 (138 days of follow-up). We included vaccination with any approved COVID-19 vaccine, and analysed first-dose vaccination only. We compared inequalities between COVID-19 and influenza vaccine uptake adjusting by age group and clinical risk, and used subgroup analysis to identify populations where inequalities were widest. The majority of individuals (871,231; 79.24%) were White British. The largest minority ethnic groups were Pakistani (50,268; 4.75%), 'other White background' (43,195; 3.93%), 'other ethnic group' (34,568; 3.14%), and Black African (18,802; 1.71%). In total, 83.64% (919,636/1,099,503) of eligible individuals received a COVID-19 vaccine. Uptake was lower compared to the White British group for 15 of 16 minority ethnic groups, with particularly wide inequalities amongst the groups 'other Black background' (hazard ratio [HR] 0.42, 95% CI 0.40 to 0.44), Black African (HR 0.43, 95% CI 0.42 to 0.44), Arab (HR 0.43, 95% CI 0.40 to 0.48), and Black Caribbean (HR 0.43, 95% CI

national Control of Patient Information (COPI) notice, which gives NHS organisations a legal requirement to share data for the purposes of the COVID-19 response. In addition, a strict governance process involving stakeholders groups (data controllers, healthcare professionals, patients and members of the public, and researchers) exists for granting researchers access to Greater Manchester Care Record data. For further details please see https://gmwearebettertogether.com/gm-care-record/ or contact GMCR-ops@manchester.ac.uk.

**Funding:** This work was funded by an internal grant from the University of Manchester. University of Manchester website: https://www.manchester.ac.uk/ The time of RW was funded by the National Institute for Health Research (NIHR) Greater Manchester Patient Safety Translational Research Centre (award number: PSTRC-2016-003). The funders had no role in study design, data collection and analysis, decision to publish, or preparation of the manuscript.

**Competing interests:** I have read the journal's policy and the authors of this manuscript have the following competing interests: MS is an NIHR investigator. The other authors have declared no competing interests.

**Abbreviations:** GM, Greater Manchester; GMCR, Greater Manchester Care Record; GP, general practice; HR, hazard ratio; IMD, Index of Multiple Deprivation; JCVI, Joint Committee on Vaccination and Immunisation; LSOA, Lower Layer Super Output Area; NHS, National Health Service; PCIE, public and community involvement and engagement.

0.42 to 0.45). In total, 55.71% (419,314/752,715) of eligible individuals took up influenza vaccination. Compared to the White British group, inequalities in influenza vaccine uptake were widest amongst the groups 'White and Black Caribbean' (HR 0.63, 95% CI 0.58 to 0.68) and 'White and Black African' (HR 0.67, 95% CI 0.63 to 0.72). In contrast, uptake was slightly higher than the White British group amongst the groups 'other ethnic group' (HR 1.11, 95% CI 1.09 to 1.12) and Bangladeshi (HR 1.08, 95% CI 1.05 to 1.11). Overall, ethnic inequalities in vaccine uptake were wider for COVID-19 than influenza vaccination for 15 of 16 minority ethnic groups. COVID-19 vaccine uptake inequalities also existed amongst individuals who previously took up influenza vaccination. Ethnic inequalities in COVID-19 vaccine uptake were concentrated amongst older and extremely clinically vulnerable adults, and the most income-deprived. A limitation of this study is the focus on uptake of the first dose of COVID-19 vaccination, rather than full COVID-19 vaccination.

## Conclusions

Ethnic inequalities in COVID-19 vaccine uptake exceeded those for influenza vaccine uptake, existed amongst those recently vaccinated against influenza, and were widest amongst those with greatest COVID-19 risk. This suggests the COVID-19 vaccination programme has created additional and different inequalities beyond pre-existing health inequalities. We suggest that further research and policy action is needed to understand and remove barriers to vaccine uptake, and to build trust and confidence amongst minority ethnic communities.

## Author summary

### Why was this study done?

- Previous research has found wide disparities in early COVID-19 vaccine uptake between ethnic groups in many countries.

- Uptake of COVID-19 vaccination was particularly low amongst older adults belonging to Black or Black British ethnic groups in the UK.

- However, previous research has tended to use broad ethnic groupings, and ethnic inequalities in COVID-19 vaccination uptake had not been contextualised by comparison with inequalities in uptake of previous vaccination programmes.

### What did the researchers do and find?

- We used electronic health records for the population of Greater Manchester, England, to estimate inequalities in COVID-19 and seasonal influenza vaccine uptake between the White British group and 16 minority ethnic groups.

- We found that ethnic inequalities in COVID-19 vaccine uptake are far wider than those seen previously for seasonal influenza vaccine uptake, and exist even amongst those recently vaccinated against influenza.

- Ethnic inequalities in COVID-19 vaccine uptake are concentrated amongst the most vulnerable—those living in the most deprived neighbourhoods, and older and extremely clinically vulnerable adults.

**What do these findings mean?**

- These findings suggest that the COVID-19 vaccine rollout has exacerbated pre-existing health inequalities in vaccine uptake.

- Themes raised in our public and community discussion groups suggest that both lower trust in COVID-19 vaccines and practical barriers to vaccine access likely contribute to lower COVID-19 vaccine uptake amongst minority ethnic groups.

- Further research and community engagement is needed to build trust and confidence amongst minority ethnic communities, and to better understand and remove barriers to vaccine access.

## Introduction

The COVID-19 pandemic has disproportionately affected people with pre-existing socioeconomic and health disadvantages, with wide inequalities in COVID-19 mortality driven by socially patterned risk factors for both SARS-CoV-2 infection and subsequent severe COVID-19 disease [1]. Older adults from minority ethnic groups have been particularly at risk, with COVID-19 mortality rates in England more than 3-fold higher amongst people belonging to Black ethnic groups, and more than 2-fold higher amongst those from Asian ethnic groups, compared to rates amongst those from White ethnic groups [2]. These disproportionate impacts are likely due in part to socioeconomic disadvantages and wide existing health inequalities for those belonging to minority ethnic groups compared to White British adults in England [2,3].

The development and rollout under emergency use legislation of several safe and effective vaccines against COVID-19 has offered the potential to protect individuals and communities from severe COVID-19 [4]. COVID-19 vaccination has been offered free at the point of use through the English National Health Service (NHS) to priority groups, based on age and clinical vulnerability to COVID-19 [5]. However, even for healthcare services provided free at the point of use, disadvantaged populations with the greatest 'need' for care tend to have lower uptake of services, formalised as the 'inverse care law' [6]. Potential barriers to vaccine uptake include logistical and practical barriers, such as the need to book and travel to vaccination appointments, as well as informational barriers [7]. Personal, family, and cultural views may also influence individuals' intention to take up vaccination, as may concerns about vaccine safety.

Access to COVID-19 vaccination initially required an appointment booked in advance via the NHS national booking service, a predominantly online booking service available only in English and that requires users to be registered with a general practice (GP) surgery [8]. Many people booked appointments following public announcements indicating that their priority group was eligible to book, and some individuals received letters in the post and/or text

messages or telephone calls from their GP inviting them to book an appointment through the national booking service. During the first few months of the COVID-19 vaccine rollout, the majority of vaccination appointments available were at out-of-town mass vaccination centres or hospital hubs. However, as the rollout has progressed, vaccinations have increasingly been provided at local and pop-up sites and without the need for pre-booked appointments, and information is increasingly (though not consistently) available in additional non-English languages.

Existing evidence from England indicates early COVID-19 vaccine uptake has been lower amongst minority ethnic groups in older adults and healthcare workers [9–14]. However, these studies focused on specific population groups, lacked granularity in ethnic group definitions, and did not compare inequalities in uptake to those of other vaccination programmes. Here, we estimate ethnic inequalities in COVID-19 vaccine uptake using a novel electronic health record dataset across a large, ethnically diverse city region in England (Greater Manchester) [15]. We analyse vaccine uptake amongst adults aged 50+ years or with clinical vulnerability to severe COVID-19, and undertake subgroup analysis to identify populations where inequalities are concentrated. To contextualise inequalities in COVID-19 vaccine uptake, we also estimate inequalities in uptake of influenza vaccine in winter 2019/2020, the season immediately prior to the major impacts of the COVID-19 pandemic. We use this comparison because similar groups of adults to those eligible for priority COVID-19 vaccination are routinely encouraged to take up seasonal influenza vaccination [16], and so this comparison allows us to see if the inequalities in vaccination uptake existed before.

## Methods

### Data collection

We extracted data on individuals aged 18+ years from the Greater Manchester Care Record (GMCR) on 18 April 2021. The GMCR is supported by Graphnet, and holds de-identified primary care and COVID-19-related electronic health record data on approximately 2.8 million patients registered with GP surgeries in Greater Manchester (GM), independent of residence or citizenship status [15]. This gives almost population-level coverage, with data from 433 of 435 (99.5%) GP surgeries in the region. The 2 GP surgeries that do not contribute data have chosen to opt out of data sharing into the GMCR. For reference, one is located in Tameside and the other is in Bolton.

### Variables

**Outcomes.**   We included data on date of first-dose COVID-19 vaccination (all approved COVID-19 vaccine types) between 1 December 2020 and 18 April 2021, and date of seasonal influenza vaccination between 1 September 2019 and 31 March 2020 (the seasonal influenza vaccine is a single dose only). Vaccinations appear in GM residents' GMCR record whether they received vaccination within or outside of GM. We used date of death (all causes) to identify censoring events during the analysis period.

**Vaccine eligibility groups.**   The Joint Committee on Vaccination and Immunisation (JCVI) created eligibility groups for COVID-19 vaccination in the UK based on clinical risk and age. There were 2 clinical risk groups. We defined the 'high clinical risk' group as patients on the Shielded Patient List who were eligible for COVID-19 vaccination in JCVI priority group 4 [5]. The clinical risk criteria for the Shielded Patient List were published by NHS Digital and included patients with severe respiratory conditions such as cystic fibrosis and severe chronic obstructive pulmonary disease, those with immunosuppression due to health conditions or treatments, and patients undergoing kidney dialysis [17]. We defined the 'moderate

clinical risk' group as patients eligible for free NHS seasonal influenza vaccination as a result of pre-existing health conditions, to approximate those eligible for COVID-19 vaccination in JCVI priority group 6 [5,16]. We defined age groups in 5-year age bands across ages 50–79 years, plus the age group 80+ years, derived from age in years at the index date (1 February 2020). We assigned COVID-19 vaccine eligibility group (by health status or age group) by determining the earliest possible eligibility for each individual in line with JCVI criteria [5].

COVID-19 vaccination eligibility groups are relatively concordant with 2019/2020 seasonal influenza vaccine eligibility criteria, with those aged 65+ years or in clinical risk groups eligible [16]. We used data from the previous season to capture 'routine' vaccine uptake amongst older or clinically vulnerable adults. We chose not to use the 2020/2021 influenza vaccination season (concurrent with COVID-19 vaccination) for 4 reasons. First, vaccine eligibility was extended to additional age groups (50+ years) [18] because of pandemic-related concerns about hospital capacity, so for many individuals this was a non-routine invitation for vaccination. Second, the widened eligibility criteria led to initial shortages in influenza vaccine supply [18], with unknown local impacts on vaccine availability. Third, there were high-profile awareness campaigns about both influenza and COVID-19 vaccine uptake that may have altered people's intentions to get vaccinated. Finally, the pandemic caused widespread disruption to in-person healthcare services and may have limited some people's access, or perception of access, to vaccination. Since influenza vaccine uptake analysis was for the previous season, we restricted the sample to individuals in one of the clinical risk groups or aged 66+ years at the index date (65 + years in 2019/2020). In contrast to the COVID-19 vaccination rollout, there were no formal criteria determining when subgroups within those eligible for 2019/2020 seasonal influenza vaccination were able to access vaccination.

**Exposures.** Ethnic group was assigned by Graphnet prior to data extraction, using an algorithm drawing on multiple electronic health record sources for each individual. NHS ethnic group categories were recoded to the 18 UK Census 2011 ethnic groups, in line with standard practice. We subsequently recoded the Gypsy or Irish Traveller group to 'other White background' due to very low numbers.

We obtained 2019 Index of Multiple Deprivation (IMD) income domain data from the Ministry of Housing, Communities and Local Government website [19] and linked these to individual-level data by residential Lower Layer Super Output Area (LSOA).

## Statistical analyses

We followed a prospective analysis plan that was submitted with the application to use the GMCR data. This plan is provided in S3 Analysis Plan, along with an explanation of the changes we made. The rationale for the main changes is outlined in the public and community involvement and engagement (PCIE) section below.

We restricted analysis to individuals resident in GM to avoid inclusion of individuals who may have subsequently registered with a non-GM GP, in which case records may be out of date. We further restricted analysis to individuals eligible for COVID-19 vaccination in priority cohorts (no missing data on age) and with complete data on LSOA and ethnicity (Fig A in S1 Appendix). Exploration of missing ethnicity data is provided in S2 Missing Data for Ethnic Group. We estimated the cumulative probability of vaccination over time stratified by COVID-19 vaccination eligibility group for both COVID-19 vaccination (first dose) and seasonal influenza vaccination, and present the results as Kaplan–Meier failure curves with 95% confidence intervals. We estimated percentage vaccine uptake at the end of the analysis period with logit-transformed confidence intervals, excluding those who died during follow-up.

To estimate inequalities in vaccine uptake, we estimated associations between ethnic group and time to vaccination (first dose only for COVID-19 vaccination) using Cox proportional hazards models for each vaccination eligibility group, treating death as a censoring event. No other form of censoring was included. It is possible a small proportion of individuals moved away from the area during the 138-day analysis period, but this is not reliably recorded in the GMCR. We used standard errors corrected for heteroscedasticity, and the Breslow method for handling ties. We used the Breslow method for pragmatic reasons as this method is much less computationally intensive than other methods, particularly for a large dataset. We confirmed that the parallel trends assumption was reasonable by inspection of log(−log[survival]) versus log(time) plots (Fig B in S1 Appendix). We also estimated associations between ethnicity and time to vaccine uptake adjusted by vaccine eligibility group (age and clinical risk groups) amongst those eligible for both COVID-19 vaccination and 2019/2020 seasonal influenza vaccination. We repeated the adjusted analysis for both COVID-19 and influenza vaccine uptake amongst this population stratified by gender. We also re-estimated results first restricting analysis to individuals resident in the least income-deprived LSOAs (IMD income domain quintile 1), then restricting the population to those resident in the most income-deprived LSOAs (quintile 5). Finally, we estimated adjusted associations between ethnic group and time to COVID-19 vaccination stratified by whether individuals previously received the 2019/2020 seasonal influenza vaccine. All results from Cox regression models are presented as hazard ratios (HRs) with 95% confidence intervals.

We estimated vaccination coverage at the end of the analysis period standardised by eligibility group using direct standardisation to the total eligible population for each vaccine, excluding individuals who died during follow-up. Results are reported by ethnic group with 95% confidence intervals.

We conducted all statistical analysis using Stata 16.1.

## Sensitivity analyses

The GMCR dataset only includes individuals who were alive when active data feeds to the GMCR from each Clinical Commissioning Group (CCG) were established. This means some individuals who died prior to the index date (1 February 2020)—and therefore potentially during the influenza vaccine analysis period—were excluded from the dataset. To assess the sensitivity of the COVID-19 and influenza vaccination comparisons to this, inequalities in COVID-19 vaccine uptake were re-estimated excluding individuals who died during follow-up. We also repeated the main analysis adjusting by income deprivation quintile and then GM locality to assess sensitivity to geographical variation. Sensitivity to missing ethnicity data is explored in S2 Missing Data for Ethnic Group.

## Public and community involvement and engagement

We held 3 online public discussion groups with diverse members of the GM community, in partnership with the NIHR Applied Research Collaboration for Greater Manchester [20] and Health Innovation Manchester [21], to inform study design. Themes and priorities raised led to our focus on inequalities in vaccine uptake (as opposed to inequalities in vaccine eligibility) and prompted comparison with seasonal influenza vaccine uptake. Key themes raised included mistrust (rooted in experiences of racism and historical context, but compounded by the pandemic); safety concerns; the complexities of personal, family, and cultural attitudes towards vaccination; and logistical, practical, and informational barriers to vaccine access. These discussions also shaped our use of language, for example our decision not to refer to vaccine 'hesitancy', as this term could be perceived as dismissing genuine concerns or delegitimising

critical dialogue. We held follow-up meetings with an advisory group formed from discussion group attendees, consisting of 4 public contributors from 4 ethnic and faith communities. We discussed preliminary results and framing of findings with the advisory group and local stake-holders including vaccination leads, with these meetings providing a means to reflect on findings and enable critical dialogue to inform subsequent work.

### Ethical approval

The GMCR Research Governance Group approved the protocol (Ref: IDCR-RQ-25, approval date: 4 February 2021) on the basis of the national control of patient information notice, which requires NHS organisations to share data to aid the COVID-19 response. All patient data were de-identified; therefore, informed consent was not required.

This study is reported as per the Strengthening the Reporting of Observational Studies in Epidemiology (STROBE) guideline (S4 Checklist).

## Results

### Descriptive statistics

There were 2,543,124 individuals aged 18+ years, resident and registered with a GP surgery in GM, and living in the community included in the GMCR dataset; 1,255,117 individuals were eligible for COVID-19 vaccination in priority cohorts (aged 50+ years or with specified health conditions). Within this group we restricted the sample to 1,099,503 individuals (87.60%) with complete data (Fig A in S1 Appendix).

Most individuals were White British (871,231; 79.24%). The largest minority ethnic groups were Pakistani (50,268; 4.57%), 'other White background' (43,195; 3.93%), 'other ethnic group' (34,568; 3.14%), and Black African (18,802; 1.71%). The population was more income-deprived than the national average, with 37.59% (413,325) living in the 20% most income-deprived neighbourhoods (Table 1).

**Table 1. Baseline characteristics and vaccine uptake.**

| Characteristic | Total analysis population[1] | Influenza vaccine eligible population[2] |
|---|---|---|
| *N* | 1,099,503 | 752,715 |
| **Ethnic group** | | |
| **White** | | |
| White British | 871,231 (79.24%) | 591,695 (78.61%) |
| White Irish | 11,877 (1.08%) | 9,287 (1.23%) |
| Other White background | 43,195 (3.93%) | 27,224 (3.62%) |
| **Mixed or multiple ethnic groups** | | |
| White and Black Caribbean | 2,760 (0.25%) | 1,893 (0.25%) |
| White and Black African | 2,752 (0.25%) | 1,830 (0.24%) |
| White and Asian | 1,848 (0.17%) | 1,266 (0.17%) |
| Other mixed or multiple background | 3,729 (0.34%) | 2,421 (0.32%) |
| **Asian or Asian British** | | |
| Indian | 17,562 (1.60%) | 12,629 (1.68%) |
| Pakistani | 50,268 (4.57%) | 39,175 (5.20%) |
| Bangladeshi | 9,560 (0.87%) | 7,975 (1.06%) |
| Chinese | 6,396 (0.58%) | 3,417 (0.45%) |

(*Continued*)

**Table 1.** (Continued)

| Characteristic | Total analysis population[1] | Influenza vaccine eligible population[2] |
|---|---|---|
| Other Asian background | 12,527 (1.14%) | 8,047 (1.07%) |
| **Black, African, Caribbean, or Black British** | | |
| Black African | 18,802 (1.71%) | 12,472 (1.66%) |
| Black Caribbean | 7,409 (0.67%) | 5,023 (0.67%) |
| Other Black background | 3,605 (0.33%) | 2,288 (0.30%) |
| **Other ethnic group** | | |
| Arab | 1,414 (0.13%) | 925 (0.12%) |
| Other ethnic group | 34,568 (3.14%) | 25,148 (3.34%) |
| **Vaccine eligibility group[3]** | | |
| **High clinical risk** | 109,267 (9.94%) | 109,267 (14.52%) |
| **Moderate clinical risk** | 253,158 (23.02%) | 253,158 (33.63%) |
| **Age (years)** | | |
| 80+ | 102,530 (9.33%) | 102,530 (13.62%) |
| 75–79 | 81,890 (7.45%) | 81,890 (10.88%) |
| 70–74 | 116,748 (10.62%) | 116,748 (15.51%) |
| 65–69 | 102,756 (9.35%) | 89,122 (11.84%) |
| 60–64 | 85,233 (7.75%) | — |
| 55–59 | 114,777 (10.44%) | — |
| 50–54 | 133,144 (12.11%) | — |
| **Received first-dose COVID-19 vaccine** | 919,636 (83.64%) | 635,670 (84.45%) |
| **Received 2019/2020 seasonal influenza vaccine** | 447,782 (40.73%) | 419,314 (55.71%) |
| **Died during follow-up period** | 26,400 (2.40%) | 24,667 (3.28%) |
| **IMD income deprivation quintile** | | |
| Least deprived | 140,374 (12.77%) | 102,078 (13.56%) |
| Q2 | 172,752 (15.71%) | 110,311 (14.66%) |
| Q3 | 155,911 (14.18%) | 117,326 (15.59%) |
| Q4 | 217,141 (19.75%) | 137,629 (18.28%) |
| Most deprived | 413,325 (37.59%) | 285,371 (37.91%) |

[1]Adults aged 50+ years or in high or moderate COVID-19 clinical risk groups.

[2]Adults aged 65+ years or in high or moderate COVID-19 clinical risk groups.

[3]Individuals were assigned to the highest priority vaccine eligibility group for which they were eligible. Priority proceeded from those aged 80+ years down the age brackets, with the high clinical risk group eligible at the same time as the age group 70–74 years, and the moderate clinical risk group eligible between the age groups 65–69 and 60–64 years. Vaccine eligibility groups are mutually exclusive, such that, for example, an individual aged 80 with high clinical vulnerability was categorized in the age 80+ eligibility group, whereas an individual aged 65 with high clinical vulnerability was categorized in the high clinical risk eligibility group.

Rapid rollout of vaccination to priority groups occurred over the 138-day analysis period (1 December 2020–18 April 2021) (Fig C in S1 Appendix), with high uptake (>76%) by the end of the period amongst all priority groups (Table A in S1 Appendix). Uptake was highest amongst those aged 75–79 years (89.34%, 95% CI 89.14% to 89.55%), with uptake decreasing monotonically across younger age groups to 76.20% (95% CI 75.99% to 76.41%) in those aged 50–54 years. Amongst those eligible due to health conditions, uptake was higher for those at high clinical risk (87.21%, 95% CI 87.02% to 87.40%) than for those at moderate clinical risk

(80.02%, 95% CI 79.87% to 80.17%). COVID-19 vaccine uptake was slightly higher amongst women (83.24%, 95% CI 83.15% to 83.33%) than men (81.70%, 95% CI 81.60% to 81.80%).

Overall, 2019/2020 seasonal influenza vaccine uptake was lower than for COVID-19 vaccination, though patterns across COVID-19 vaccination eligibility groups were similar to those for COVID-19 vaccine uptake (Table A and Fig C in S1 Appendix). Uptake was highest amongst those aged 80+ years (70.18%, 95% CI 69.90% to 70.46%), and lowest amongst those eligible due to moderate clinical risk (36.82%, 95% CI 36.64% to 37.00%). Influenza vaccine uptake was slightly lower amongst women (52.93%, 95% CI 52.78% to 53.08%) compared to men (53.78%, 95% CI 53.62% to 53.94%).

### Ethnic inequalities in COVID-19 vaccine uptake

Inequalities in COVID-19 vaccine uptake between the White British group and minority ethnic groups were large across all eligibility groups (Fig 1A; Table B in S1 Appendix). Compared to the White British group, uptake differences tended to be largest for the ethnic groups Black African (HR range: 0.28 [95% CI 0.24 to 0.32] to 0.47 [95% CI 0.45 to 0.49]), Black Caribbean (HR range: 0.36 [95% CI 0.33 to 0.38] to 0.57 [95% CI 0.51 to 0.64]), 'other Black background' (HR range: 0.38 [95% CI 0.29 to 0.50] to 0.48 [95% CI 0.40 to 0.57]), and Arab (HR range: 0.38 [95% CI 0.30 to 0.49] to 0.54 [95% CI 0.40 to 0.74]). Patterns of inequalities in uptake across eligibility groups varied between ethnic groups, decreasing with decreasing age for some (6 of 16) groups (e.g., Black African and Indian) but increasing amongst younger age groups for others (e.g., 'other White background' and 'other ethnic group'). For 5 of 16 minority ethnic groups, the widest inequality compared to the White British group occurred within the high clinical risk eligibility group. In line with these results, there were substantial differences in standardised total vaccine uptake at the end of the analysis period, with eligibility-group-standardised uptake at least 20 percentage points lower than that of the White British group for 6 of 16 minority ethnic groups (Table C in S1 Appendix).

### Ethnic inequalities in influenza vaccine uptake

There were some moderate inequalities in influenza vaccine uptake by ethnic group (Fig 1B; Table D in S1 Appendix). Considering each vaccine eligibility group, inequalities compared to the White British group were widest for the groups White and Black African (HR range: 0.45 [95% CI 0.28 to 0.75] to 0.76 [95% CI 0.67 to 0.85]) and Black African (HR range: 0.53 [95% CI 0.46 to 0.62] to 0.77 [95% CI 0.47 to 0.80]). Following adjustment by eligibility group, overall inequalities compared to the White British group in influenza vaccine uptake were widest for the groups White and Black Caribbean (HR 0.63, 95% CI 0.58 to 0.68) and White and Black African (HR 0.67, 95% CI 0.63 to 0.72) (Fig 1C; Table E in S1 Appendix). In contrast, relative time to vaccination was shorter in at least 1 eligibility group for the ethnic groups Indian, Pakistani, Bangladeshi, and 'other ethnic group' compared to the White British group (Fig 1B; Table D in S1 Appendix). Similarly, influenza vaccine uptake at the end of the analysis period (standardised by vaccine eligibility group) was over 3 percentage points higher for the ethnic groups Bangladeshi and 'other ethnic group' compared to the White British group (Table C in S1 Appendix).

### Comparison of vaccine uptake inequalities

Amongst the population eligible for both vaccinations, inequalities in COVID-19 vaccine uptake adjusted by eligibility group exceeded inequalities in influenza vaccine uptake for almost all (15 of 16) minority ethnic groups (Fig 1C; Table E in S1 Appendix). The magnitudes of the differences were substantial, with inequalities in COVID-19 vaccine uptake at least

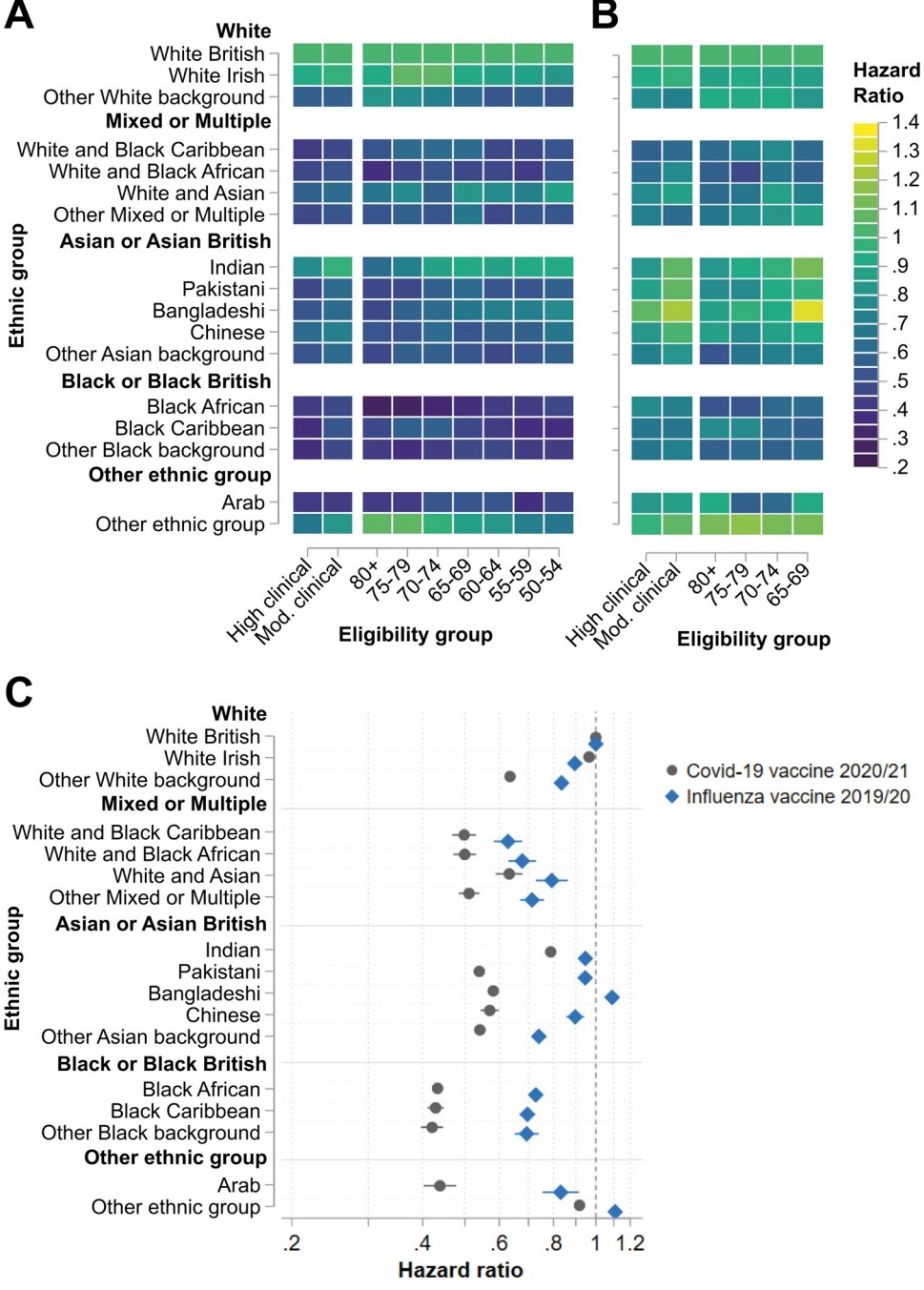

**Fig 1. Associations between ethnic group and vaccine uptake.** Hazard ratios with 95% confidence intervals from Cox proportional hazards models estimating time to vaccination across ethnic groups stratified by vaccine eligibility group for (A) COVID-19 vaccine and (B) 2019/2020 seasonal influenza vaccine. (C) Hazard ratios with 95% confidence intervals from Cox proportional hazards models estimating time to vaccination across ethnic groups adjusted by vaccine eligibility group amongst individuals eligible for both the COVID-19 and 2019/2020 seasonal influenza vaccines. High clinical = high clinical risk. Mod. clinical = moderate clinical risk.

double those for influenza vaccine uptake for 7 of 16 minority ethnic groups. Inequalities in vaccine uptake between ethnic groups followed very consistent patterns amongst men and women, though the magnitudes of inequalities were moderately greater for women than men

for many minority ethnic groups (10 of 16 for COVID-19 vaccination; 7 of 16 for influenza vaccination) (Fig D and Table F in S1 Appendix). There were no minority ethnic groups for whom inequalities in vaccine uptake were statistically significantly greater amongst men than women.

Results were not sensitive to excluding those who died during the COVID-19 vaccination follow-up period, to parallel exclusion in influenza vaccine analysis (Table E in S1 Appendix), and were robust to different handling of missing ethnicity data (S2 Missing Data for Ethnic Group). Results were also not sensitive to adjustment by neighbourhood-level income deprivation or geographical locality (Fig E and Table G in S1 Appendix).

## Subgroup analysis by income deprivation

Uptake of COVID-19 and influenza vaccines followed the income deprivation gradient, with uptake of each vaccine approximately 9 percentage points lower in the most, compared to the least, income-deprived areas (Table A in S1 Appendix). Restricting the population eligible for both vaccinations to those living in the least income-deprived areas attenuated the differences in ethnic inequalities in COVID-19 vaccine uptake as compared to those for influenza vaccine uptake, with inequalities greater for COVID-19 vaccine uptake in only 4 of 16 minority ethnic groups (Fig 2A; Table H in S1 Appendix). In contrast, wide ethnic inequalities in COVID-19 vaccine uptake persisted within the most income-deprived areas, exceeding the magnitude of influenza vaccine uptake inequalities for almost all (15 of 16) minority ethnic groups (Fig 2B; Table H in S1 Appendix).

## Subgroup analysis by prior influenza vaccine uptake

Amongst those eligible for both vaccinations, COVID-19 vaccine uptake was substantially higher amongst those who did previously take the influenza vaccine (95.46%, 95% CI 95.40%

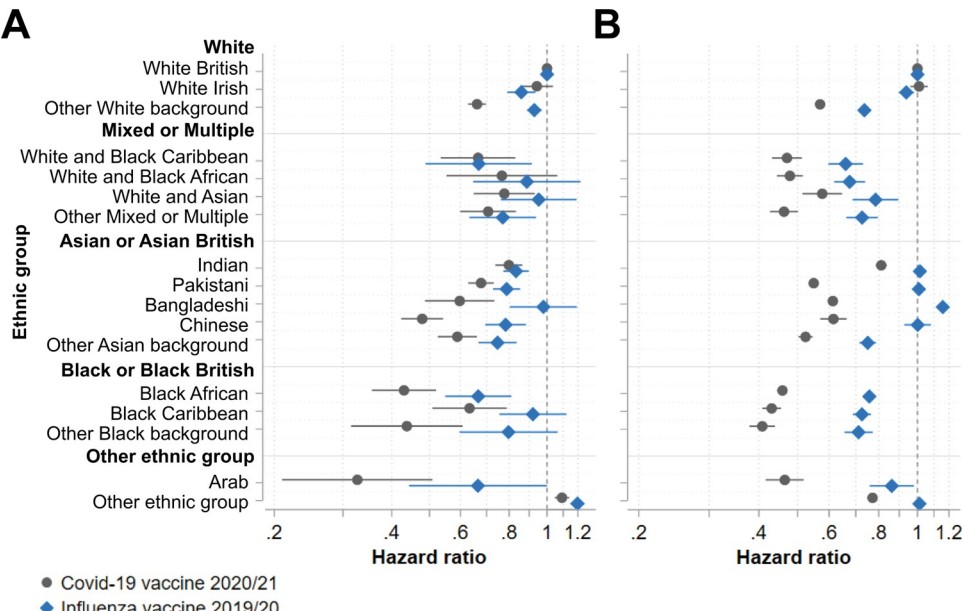

**Fig 2. Associations between ethnic group and vaccine uptake by income deprivation.** Hazard ratios with 95% confidence intervals from Cox proportional hazards models estimating time to vaccination across ethnic groups adjusted by vaccine eligibility group amongst individuals eligible for both the COVID-19 and 2019/2020 seasonal influenza vaccines living in (A) the 20% least income-deprived neighbourhoods and (B) the 20% most income-deprived neighbourhoods.

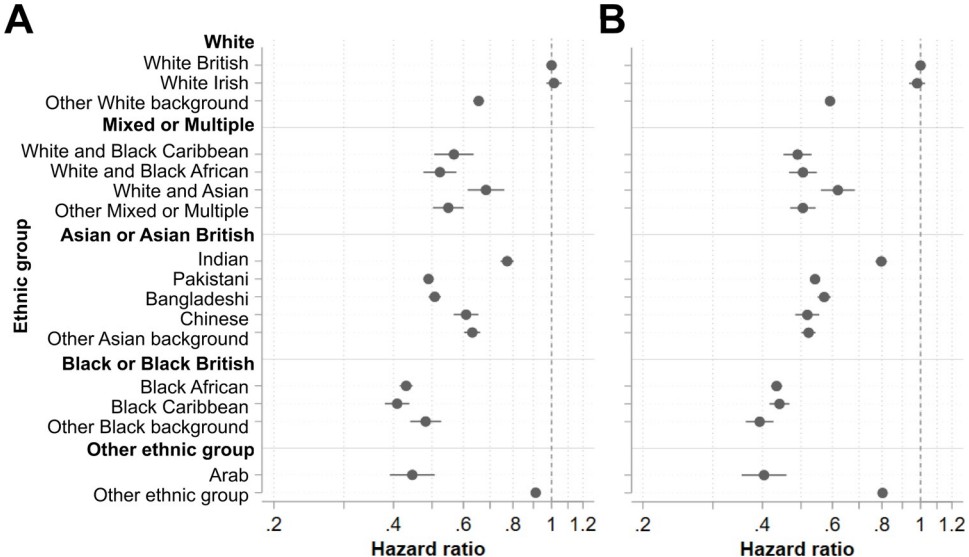

**Fig 3. Associations between ethnic group and vaccine uptake by prior influenza vaccine uptake.** Hazard ratios with 95% confidence intervals from Cox proportional hazards models estimating time to COVID-19 vaccination across ethnic groups adjusted by vaccine eligibility group amongst individuals eligible for both the COVID-19 and 2019/2020 seasonal influenza vaccines who (A) did take up 2019/2020 seasonal influenza vaccination and (B) did not take up 2019/2020 seasonal influenza vaccination.

to 95.52%) than those who did not (74.64%, 95% CI 74.55% to 74.74%) (Table A in S1 Appendix). Amongst those who did previously take the influenza vaccine, wide ethnic inequalities in COVID-19 vaccine uptake persisted (15 of 16 groups) and were similar in magnitude to the inequalities amongst those not previously vaccinated against influenza for half (8 of 16) of the minority ethnic groups (Fig 3; Table I in S1 Appendix). Where there were differences, these were relatively small in magnitude and varied in directionality.

## Discussion

We found wide inequalities in time to COVID-19 vaccination for most minority ethnic groups compared to the White British group. Inequalities were particularly wide for all Black or Black British groups and the Arab group. Vaccine uptake inequalities were substantially wider for COVID-19 vaccination than for influenza vaccination. We also found wide ethnic inequalities in COVID-19 vaccine uptake amongst individuals who previously took up influenza vaccination. COVID-19 vaccine inequalities were concentrated amongst those most at risk of severe COVID-19 (older and clinically vulnerable adults) and those living in income-deprived neighbourhoods.

Our finding that COVID-19 vaccine uptake was lower amongst almost all minority ethnic groups, and particularly low amongst people belonging to Black or Black British ethnic groups, is broadly consistent with existing reports [9–14]. However, previous studies have not analysed uptake inequalities for Arab individuals, so these inequalities were not previously recognised [9–14].

To the best of our knowledge, there is no published information about ethnic inequalities in influenza vaccine uptake amongst older or clinically vulnerable adults in the UK. However, several studies assessing intention to receive vaccination against the 2009 pandemic H1N1 influenza ('swine flu') found that people from minority ethnic groups were more likely than the White British group to intend to take up vaccination [22]. It has been suggested this could

be attributable to the higher incidence of severe influenza amongst the minority ethnic population [22]. However, this factor appears not to apply to COVID-19 vaccination. These findings reinforce the conclusion that the magnitude of ethnic inequality in COVID-19 vaccine uptake is unusual, and far exceeds the inequalities associated with uptake of other vaccines.

Recent prior influenza vaccine uptake suggests individuals have a degree of confidence in vaccination, and have chosen and successfully navigated access to vaccination, perhaps somewhat explaining the higher COVID-19 vaccine uptake amongst this group. However, even within the group with recent influenza vaccine uptake, we found lower uptake of COVID-19 vaccination amongst people from minority ethnic groups relative to the White British group. This suggests that new ethnic health inequalities have been generated, perhaps resulting from additional barriers to vaccine access, lower vaccine confidence, or both.

There were many differences in how individuals were able to access COVID-19 vaccination compared to 2019/2020 influenza vaccination, which may have introduced additional barriers that disproportionately affected minority ethnic groups. For example, there was a strict requirement for booked COVID-19 vaccination appointments, and much delivery was through mass vaccination centres or hospital hubs. As discussed by Razai et al. [7], residential segregation driven by systemic racism may therefore have resulted in additional barriers—in terms of journey time and cost—to accessing centralised vaccination sites for some communities. In contrast, influenza vaccines are available through GPs and community pharmacies, often via drop-in services [16]. Community pharmacies have been shown to be more accessible in areas of high deprivation than in less deprived areas [23]. Participants in PCIE discussion groups also raised concerns that vaccine information was rarely presented in languages other than English, and that public health messaging sometimes failed to address the concerns of target populations. It is unclear whether any type of health equity impact assessment or community engagement was undertaken during the very rapid centralised design of the COVID-19 vaccine rollout, and this may have contributed to the observed exacerbation of inequalities. Further interdisciplinary research is needed to understand the potential impacts of each of these factors on equity of access to COVID-19 vaccination.

Several studies have investigated attitudes towards COVID-19 vaccination across sociodemographic groups, with relatively consistent reports of lower vaccine confidence amongst minority ethnic groups, which likely contributes to observed uptake disparities [24–29]. Concerns about the novelty and potential unknown side effects or long-term effects of COVID-19 vaccines were frequently reported [24,26–28]. These concerns may have been heightened amongst people from minority ethnic groups due to their underrepresentation in COVID-19 vaccine clinical trials [7,26,29]. Erosion of trust in government, mainstream media, and medical authorities during the COVID-19 pandemic has also been reported [7,25–31] and was a key theme raised in PCIE discussions. A combination of factors—such as the disproportionate burden of COVID-19 mortality amongst minority ethnic groups [2], apparent government suppression of a report into COVID-19 disparities [30], and disregard for the impact of restrictions on non-Christian religious festivals—may have exacerbated loss of trust amongst minority ethnic groups [7]. Moreover, these factors likely compounded existing mistrust amongst some minority ethnic communities stemming from racism, experiences of culturally insensitive healthcare, and awareness of previous unethical healthcare research [7,24]. It will be particularly important to understand whether these COVID-19 vaccination findings represent a shift in attitudes towards vaccination per se, as this may lead to wider inequalities in future routine influenza and other vaccination campaigns.

A strength of this study is the comparison of inequalities in uptake of COVID-19 and influenza vaccines. This brings crucial context to observed inequalities, and demonstrates that COVID-19 vaccine uptake reproduces previous health inequalities, but also harbours new

inequalities that are particularly concentrated in income-deprived neighbourhoods. Stratification by previous influenza vaccine uptake demonstrated that the COVID-19 vaccine rollout has generated inequalities even amongst individuals previously accessing influenza vaccination.

Another strength is the population-level coverage of electronic health record data across a large, ethnically diverse metropolitan area, which allowed estimation of inequalities for 16 minority ethnic groups. However even with this large sample, we were unable to estimate inequalities in vaccine uptake for the Gypsy or Irish Traveller group due to very low numbers. There are also limitations in the recording of sociodemographic information in electronic health records. For example, there is incomplete granularity in the recording of ethnicity, which leads to relatively large populations in each of the 'other' ethnic groupings, a problem highlighted by our community advisory group. Analysis stratified by gender indicated that ethnic inequalities in vaccine uptake were moderately wider for women than for men for many minority ethnic groups. This highlights the interactions between sociodemographic factors that may exacerbate health inequalities amongst some groups. However, we were unable to explore the role of other factors that may intersect with ethnicity and affect vaccination uptake, such as faith, country of birth, or preferred spoken and written languages, as these are rarely recorded in electronic health records.

The high clinical risk group was robustly identified using clinical codes for the 'extremely clinically vulnerable' population of individuals who were instructed to 'shield' during the first COVID-19 wave in England. However, the moderate clinical risk group was approximated using codes for clinical eligibility for seasonal influenza vaccination, potentially introducing some misclassification. Similarly, health and social care workers were not identified in our dataset but were eligible for priority vaccination. These misclassifications may have differed with respect to ethnicity, but Kaplan–Meier plots indicated reasonable temporal agreement between eligibility and uptake, suggesting this is unlikely to have substantially biased the estimates. Data were highly complete ($\geq$99.9%) for most variables, but 12.31% of the population was missing data for ethnic group, which could have introduced bias. Detailed exploration of the demographic patterns of missing ethnicity data suggested that a large majority of those with missing data were likely to be White British. Sensitivity analysis showed this did not substantially bias our estimates of vaccine uptake inequalities. Another limitation of our analysis is that we used the Breslow method to handle ties in the Cox regression models, as opposed to the Efron method, which has been shown to generally be less biased [32]. Although using the Breslow method may have biased our results towards underestimation of inequalities in vaccine uptake, this method is more computationally feasible with a large dataset and is unlikely to have markedly biased estimates, given our large sample size [32].

Another limitation is that we considered only first-dose COVID-19 vaccine uptake. Reported differential uptake of second doses [14] may already have led to wider inequalities in terms of full COVID-19 vaccination than those reported here. It may be particularly important to understand whether inequalities widen for subsequent doses, given the planned rollout of 'booster' COVID-19 vaccine doses [33].

Finally, given the ethnically diverse urban study setting, our results are likely informative for similar regions, but may not be fully generalizable to England as a whole, or applicable across international contexts.

In summary, we identified wide ethnic inequalities in COVID-19 vaccine uptake, and demonstrated that inequalities were concentrated amongst groups at greatest risk of severe COVID-19—older adults, extremely clinically vulnerable individuals, and those living in the most income-deprived neighbourhoods. By contextualising inequalities in relation to previous influenza vaccine uptake, we found a marked exacerbation of ethnic health inequalities in

vaccine uptake. Further research and policy action is needed to understand and remove barriers to vaccine uptake, and to build trust and confidence amongst minority ethnic communities.

## Supporting information

**S1 Appendix. Supplementary figures and tables.** Fig A: Flow chart showing inclusion criteria and missing data. GM, Greater Manchester; GP, general practice; LSOA, Lower Layer Super Output Area. LSOAs are neighbourhood-level administrative boundaries containing approximately 1,000 residents. Fig B: log(−log[survival]) versus log(time) plots stratified by ethnic group for COVID-19 and influenza vaccine uptake. Fig C: Kaplan–Meier failure curves indicating the cumulative probability of vaccination over time for (A) COVID-19 and (B) influenza vaccines, stratified by COVID-19 vaccination eligibility group. Individuals were assigned to the highest priority vaccine eligibility group for which they were eligible. Priority proceeded from those aged 80+ years down the age brackets, with the high clinical risk group eligible at the same time as the age group 70–74 years, and the moderate clinical risk group eligible between the age groups 65–69 and 60–64 years, as indicated in the legend group order. Vaccine eligibility groups are mutually exclusive, such that, for example, an individual aged 80 with high clinical vulnerability was categorized in the age 80+ eligibility group, whereas an individual aged 65 with high clinical vulnerability was categorized in the high clinical risk eligibility group. Fig D: Associations between ethnic group and vaccine uptake by gender (results also in Table F). Hazard ratios with 95% confidence intervals from Cox proportional hazards models estimating time to vaccination across ethnic groups, adjusted by vaccine eligibility group, stratified by gender. (A) Male; (B) female. Fig E: Associations between ethnic group and COVID-19 vaccine uptake—sensitivity analysis adjusting by locality or income deprivation (results also in Table G). Hazard ratios with 95% confidence intervals from Cox proportional hazards models estimating time to COVID-19 vaccination across ethnic groups, adjusted by vaccine eligibility group, plus additional adjustment by GM locality (10 local authority areas) or income domain quintile. Table A: Vaccine uptake by population subgroup and vaccine type (percentage uptake and 95% CI). Table B: Associations between ethnic group and COVID-19 vaccine uptake (results also in Fig 1A). Hazard ratios with 95% confidence intervals from Cox proportional hazards models estimating time to COVID-19 vaccination across ethnic groups, stratified by vaccine eligibility group. Table C: Vaccine uptake standardised to GM vaccine eligibility group structure. Estimated percentage uptake with 95% confidence intervals. Table D: Associations between ethnic group and influenza vaccine uptake (results also in Fig 1B). Hazard ratios with 95% confidence intervals from Cox proportional hazards models estimating time to influenza vaccination across ethnic groups, stratified by vaccine eligibility group. Table E: Associations between ethnic group and vaccine uptake (results also in Fig 1C). Hazard ratios with 95% confidence intervals from Cox proportional hazards models estimating time to vaccination across ethnic groups, adjusted by vaccine eligibility group. Table F: Associations between ethnic group and vaccine uptake by gender (results also in Fig D). Hazard ratios with 95% confidence intervals from Cox proportional hazards models estimating time to vaccination across ethnic groups, adjusted by vaccine eligibility group, stratified by gender. Table G: Associations between ethnic group and COVID-19 vaccine uptake—sensitivity analysis adjusting by locality or income deprivation (results also in Fig E). Hazard ratios with 95% confidence intervals from Cox proportional hazards models estimating time to COVID-19 vaccination across ethnic groups, adjusted by vaccine eligibility group, plus additional adjustment by GM locality or income domain quintile. Table H: Associations between ethnic group and vaccine uptake by income deprivation (results also in Fig 2).

Hazard ratios with 95% confidence intervals from Cox proportional hazards models estimating time to vaccination across ethnic groups, adjusted by vaccine eligibility group, stratified by income deprivation. Table I: Associations between ethnic group and vaccine uptake by prior influenza vaccine uptake (results also in Fig 3). Hazard ratios with 95% confidence intervals from Cox proportional hazards models estimating time to COVID-19 vaccination across ethnic groups, adjusted by vaccine eligibility group, stratified by prior influenza vaccine uptake. (PDF)

**S2 Missing Data for Ethnic Group. Supplementary figures and tables regarding missing data for ethnic group.** Table A: Missing ethnicity data by population subgroup (percentage missing and 95% CI). Table B: Census estimates of the percentage of individuals from each ethnic group by GM locality. Table C: Percentage of individuals in sample population from each ethnic group by GM locality, including missing data as an additional ethnic group category. Table D: Difference between sample population and census estimates (sample population estimate minus census estimate) of percentage of individuals from each ethnic group by GM locality, including missing data as an additional ethnic group category. Table E: Associations between ethnic group and vaccine uptake following partial recoding of missing ethnicity data (results also shown in Fig B). Hazard ratios with 95% confidence intervals from Cox proportional hazards models estimating time to vaccination across ethnic groups, adjusted by vaccine eligibility group, following partial recoding of missing ethnicity data to White British. Fig A: Histogram showing the distribution of percentage missing ethnic group data across LSOAs (neighbourhoods). Fig B: Associations between ethnic group and vaccine uptake following partial recoding of missing ethnicity data (results also shown in Table E). Hazard ratios with 95% confidence intervals from Cox proportional hazards models estimating time to vaccination across ethnic groups, adjusted by vaccine eligibility group, following partial recoding of missing ethnicity data to White British, compared to original ethnicity coding used in the main analysis. (A) COVID-19 vaccination; (B) 2019/2020 seasonal influenza vaccination. (PDF)

**S3 Analysis Plan. Prospective analysis plan with information about changes made.** (PDF)

**S4 Checklist. Strengthening the Reporting of Observational Studies in Epidemiology (STROBE) checklist completed with section and paragraph numbers for each item.** (PDF)

## Acknowledgments

We would like to thank John Ainsworth and all members of the Greater Manchester Population Research Resource team at the University of Manchester for helpful discussion and suggestions that have helped shape this project. We are grateful to Nicky Timmis, Aneela McAvoy, Joanna Ferguson, and Sue Wood for their support in organising and hosting PCIE discussion groups. We would particularly like to thank Nasrine Akhtar, Basma Issa, Nicholas Filer, and Charles Kwaku-Odoi (public contributor advisory group members) for their invaluable insights, as well as all members of the NIHR Applied Research Collaboration for Greater Manchester Public and Community Involvement and Engagement Panel and the Health Innovation Manchester Public Community Involvement and Engagement Forum.

We would also like to recognise the GMCR (a partnership of Greater Manchester Health and Social Care Partnership, Health Innovation Manchester, and Graphnet Health, on behalf of Greater Manchester localities) for the provision of data required to undertake this work.

This work uses data provided by patients and collected by the NHS as part of patient care and support. Using patient data is vital to improving health and care for everyone. There is huge potential to make better use of information from people's patient records, to understand more about disease, develop new treatments, monitor safety, and plan NHS services. Patient data should be kept safe and secure, to protect everyone's privacy, and it's important that there are safeguards to make sure that data are stored and used responsibly. Everyone should be able to find out about how patient data is used.

## Author Contributions

**Conceptualization:** Ruth Elizabeth Watkinson, Stephanie Gillibrand, Caroline Sanders, Matt Sutton.

**Data curation:** Richard Williams.

**Formal analysis:** Ruth Elizabeth Watkinson.

**Funding acquisition:** Caroline Sanders, Matt Sutton.

**Investigation:** Ruth Elizabeth Watkinson, Stephanie Gillibrand, Caroline Sanders, Matt Sutton.

**Methodology:** Ruth Elizabeth Watkinson, Richard Williams, Matt Sutton.

**Resources:** Caroline Sanders, Matt Sutton.

**Supervision:** Caroline Sanders, Matt Sutton.

**Validation:** Stephanie Gillibrand.

**Visualization:** Ruth Elizabeth Watkinson, Stephanie Gillibrand, Matt Sutton.

**Writing – original draft:** Ruth Elizabeth Watkinson.

**Writing – review & editing:** Ruth Elizabeth Watkinson, Richard Williams, Stephanie Gillibrand, Caroline Sanders, Matt Sutton.

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
