## [Editor Report · Decision Letter 0]

19 Oct 2021

Dear Dr Watkinson, 

Thank you for submitting your manuscript entitled "Ethnic inequalities in Covid-19 vaccine uptake and comparison to seasonal Influenza vaccine uptake: retrospective cohort study" for consideration by PLOS Medicine.

Your manuscript has now been evaluated by the PLOS Medicine editorial staff and I am writing to let you know that we would like to send your submission out for external peer review.

Please re-submit your manuscript within two working days, i.e. by Oct 21 2021 11:59PM.

Kind regards,

Callam Davidson

Associate Editor

PLOS Medicine

---

## [Decision Letter · Decision Letter 1]

9 Dec 2021

Dear Dr. Watkinson,

Thank you very much for submitting your manuscript "Ethnic inequalities in Covid-19 vaccine uptake and comparison to seasonal Influenza vaccine uptake: retrospective cohort study" (PMEDICINE-D-21-04381R1) for consideration at PLOS Medicine. 

Your paper was evaluated by an associate editor and discussed among all the editors here. It was also discussed with an academic editor with relevant expertise, and sent to independent reviewers, including a statistical reviewer. The reviews are appended at the bottom of this email and any accompanying reviewer attachments can be seen via the link below:

[LINK]

In light of these reviews, I am afraid that we will not be able to accept the manuscript for publication in the journal in its current form, but we would like to consider a revised version that addresses the reviewers' and editors' comments. Obviously we cannot make any decision about publication until we have seen the revised manuscript and your response, and we plan to seek re-review by one or more of the reviewers. 

We hope to receive your revised manuscript by Dec 30 2021 11:59PM. Please email us (plosmedicine@plos.org) if you have any questions or concerns.

We look forward to receiving your revised manuscript. 

Sincerely,

Callam Davidson, 

PLOS Medicine

plosmedicine.org

Comments from the academic editor:

There should be some more information about how persons, who are eligible for a vaccination, get that information. Do they receive an invitation letter? And is this letter in different languages? Are there any additional efforts to reach and motivate people to make a vaccination appointment? Especially for non-UK readers, it is important to understand how the system works.

Title: Please update title to ‘Ethnic inequalities in Covid-19 vaccine update and comparison to seasonal Influenza vaccine update in Greater Manchester: a cohort study’.

Financial disclosure: Please provide more information (see here for details of what is required): https://journals.plos.org/plosmedicine/s/submission-guidelines#loc-financial-disclosure-statement

Abstract Methods and Findings:

* Please include the study design, time period during which the study took place, length of follow up, and main outcome measures.

* Please include basic demographic data (summarising that presented on lines 198 – 201).

* Please include the important dependent variables that are adjusted for in the analyses.

Line 27-8: Please update to ‘This suggests that the Covid-19 vaccination programme has created…’.

Citations should be in square brackets, and preceding punctuation.

Please remove the ‘Role of funders statement’ (Line 175) and instead update your Financial disclosure (submission form).

Please remove the ‘Competing interests statement’ (Line 188) and instead include this information in your submission form. In the event of publication, this information is included as metadata.

Please ensure that the study is reported according to the STROBE guideline, and include the completed STROBE checklist as Supporting Information. Please add the following statement, or similar, to the Methods: "This study is reported as per the Strengthening the Reporting of Observational Studies in Epidemiology (STROBE) guideline (S1 Checklist)."

Did your study have a prospective protocol or analysis plan? Please state this (either way) early in the Methods section.

Please remove subheadings from your Discussion.

Please re-order your discussion such that comparison to the literature and interpretation precedes strengths and limitations.

Please remove funding details from your acknowledgements and include them in the Financial disclosure (submission form). Specific grant numbers should be included.

Please remove the ‘Author contributions’ section (similar to comments above, this information will be included as metadata based on submission form responses in the event of acceptance).

When citing preprints (e.g. reference 13), please include [preprint] per the instructions here: https://journals.plos.org/plosmedicine/s/submission-guidelines#loc-references

Please define the abbreviations in Figure S1.

Comments from the reviewers:

Reviewer #1: This study reveals the vaccine uptake inequalities by using a reliable public dataset. The results, including sensitivity analyses and public discussion, seem to justify the authors' conclusions. Please consider the following minor comments.

#1 L35

Adding more detail about the pandemic situation in England will help me understand. How much is the COVID-19 morbidity or mortality among the older adults from minority ethnic groups compared to White British adults in England?

#2 L43, L338

How are booking or travelling to vaccination appointments difficult for ethnic minority groups in England? Is it available through the internet, telephone, or hand-writing? How far are the vaccination centres are from their residence? Is there no language support? Are the workers available to access vaccination at night or their workplace?

#3 L329

I suppose fewer opportunities existed to reflect ethnic minority groups' voices in the new-introduced vaccination system. The rapid political decision-making process might play a significant role in such an unusual inequality.

#4 L351

I think the other sociodemographic factors might be the confounders in this study. The Covid-19 vaccine uptake may also differ across the participants' years of residence, occupation, or having family members.

Is there any possibility the participants had already got vaccinated in the other area? Does the database cover the population only who has permanent citizenship?

Reviewer #2: The authors present a detailed report of COVID-19 vaccination uptake among different racial/ethnic groups from a large multicultural city in the United Kingdom. Using survival analysis in an enormous data set, they give compelling evidence of disparities in vaccination uptake between groups. They additionally show how these disparities persist when stratified by deprivation, age category and vaccine eligibility group. Throughout, they contextualise COVID-19 vaccination uptake by comparing it to influenza vaccination uptake prior to the start of the pandemic. All in all, these results suggest that existing racial/ethnic disparities in vaccination uptake were exacerbated by the pandemic. This is vital evidence to encourage health policy makers to act to address these disparities and, additionally, for health researchers to investigate the specific causes of these disparities.

I thought this was a very well-written manuscript and I appreciated the great level of detail the authors went to to present their findings, including a rigorous exploration of bias due to missing data and sensitivity analyses. A few minor questions to address follow:

Introduction

-The final sentence of the penultimate and ultimate paragraphs (i.e. the sentences starting "However, these studies have focused..." and, "We use this comparison because similar groups...") could be rephrased to make it clearer that your study is using the influenza vaccine uptake comparison as an example of a pre-pandemic disparity in routine vaccine uptake. I suggest rephrasing the first sentence I referenced like so: "..., and have not been compared to inequalities in uptake of other vaccination programmes." The final sentence in the Introduction could be rephrased to: "...this allows us to see if the inequalities in vaccination uptake existed before." As it stands, these sections make me think you want to show that, if your study had revealed no disparity in influenza vaccination uptake by racial/ethnic group, that would be evidence that there were no such health disparities at all (i.e. in vaccination programme uptake or beyond).

Methods

-Data Collection: Please report why data were unavailable for one surgery, and where it was located

-Exposures: Although the number of participants in the Gypsy or Irish Traveller group may be too small to report meaningful statistics in this study, perhaps reporting them in the supplementary materials would be helpful for potential future researchers wishing to explore vaccination patterns in this ethnic group via meta-analysis.

-Statistical analyses: The Efron method for handling ties is generally less biased than the Breslow method (see Hertz-Picciotto and Rockhill 1997: Validity and Efficiency of Approximation Methods for Tied Survival Times in Cox Regression). Why did you choose the Breslow method instead?

Results

-Ethnic inequalities in Covid-19 vaccine uptake: The way the hazard ratio ranges are reported in the text is rather difficult to read; it can be hard to disentangle the 95% confidence interval values from the point estimates. I suggest either omitting the 95% confidence interval values from the text (because they are reported in Table S2 and quite narrow anyway) or changing the format to something like: "Hazard ratio (HR) range 0.276 (95% CI: 0.241, 0.317) to 0.471 (95% CI: 0.450, 0.494)."

-I am curious to see how these results differed by gender within racial/ethnic groups. Could you please report a subgroup analysis (e.g. in the same format as Figure 2) showing vaccine uptake patterns by racial/ethnic group stratified by gender?

Reviewer #3: Alex McConnachie, Statistical Review

The paper by Watkinson et al examines variations in COVID-19 vaccination uptake in 2020-21 between ethnic groups, and compares these with variations in flu vaccination uptake in 2019-20. This review considers the use of statistics in the paper.

Overall, the statistical methods and presentation is very good. The main analyses are supported by a range of adjusted, stratified, and sensitivity analyses, and the results are not over-interpreted. My comments are quite minor.

For my taste, there are too many decimal places reported. I think one less could be used throughout the paper.

Figure 3 might work better with the two panels overlaid on the same plot.

The Kaplan-Meier plots show survival probabilities, which is not incorrect, but may be confusing to a general readership, since the paper is not about mortality. The majority of the paper talks in terms of vaccine uptake. Would it therefore be better to flip the y-axis of the K-M plots, and show the cumulative probability of vaccination?

Looking at Figure S3, I assume the order of the legends is in terms of the final vaccination probability? Perhaps this would be clearer if the probabilities were shown (with 95% CIs?) as part of the legend. Also, I'm not sure if it is my eyesight, but the colours in the legend do not seem to match the colours of the curves.

Figure S3 is quite cramped, with 3 estimates and CIs per ethnic group. Could the y axis be stretched a little?

In Figure S6, a slight vertical offset to separate each pair of estimates and CIs, as in other similar figures, would help.

The sensitivity analysis that includes geographic adjustment uses 10 LA areas, which is fine. Not for this paper, but would it be possible to do (and worth doing) an analysis by LSOA, adjusting for spatial correlation, and explore the associations between uptake and factors such as distance to nearest vaccination centre?

Reviewer #4: Many thanks for the opportunity to review this very interesting paper, which is of considerable public health relevance. I have some minor suggestions, which I think require clarification and which may improve the quality of this manuscript.

Abstract:

- It should be made clear that this paper is based on first/any dose of Covid-19 vaccination, and does not present information on full course of vaccination. This is clarified in the manuscript, but is relevant in the abstract too.

- It would be helpful to include any information on vaccine type(s)/make(s)

Methods:

- Was there any other form of censoring considered e.g. moving out of the region?

- Further information on full vaccination status would be helpful in the Methods, even if just to advise the reader that this was not possible to measure

- Some further elaboration on the "shielded patient list" would be helpful. This may be clear to a UK audience, but may be less clear for other international readers

- PPI section is excellent and a real strength of this paper

Results:

- The second paragraph of Descriptive Statistics section does not mention Pakistanis although they appear from the data to be the second largest ethnic group

- Table 1 under Vaccine eligibility group, are the clinical risk groups (high vs. moderate) and age groups mutually exclusive or is there overlap between some of these? Can this be clarified? If I understand correctly, you assigned individuals to whichever eligibility group they first fell in to, but what did this mean in practice for most older patients? Were they more likely to be in the moderate or high clinical risk groups (based on comorbidities), or more likely to be in their respective age groups? A footnote in this table may be helpful. 

- Summary data are presented in Table 1 for death during follow-up. Is there any information on what proportion of deaths might have been attributed to Covid (either as primary or contributory cause)? Moreover, any summary data on Covid infection rates may be helpful for context if available.

- The findings relating to inequalities in vaccine uptake appear to be quite robust and are important to note. It would be helpful to contextualise this further by translating some of the key results (which are all presented in relative terms using HRs etc) in to absolute numbers if possible. For this population of approx. 1.1 million people, how do these results translate in to numbers of missed vaccines for certain ethnicities or certain risk groups? Translation in to absolute numbers may be particularly helpful for public health messaging.

- Did you have any information on length of time residing in the UK? These individuals are all registered with GPs but I do not get a sense whether they are all long-standing UK citizens who are otherwise integrated in to UK society, or if they might be relatively recent arrivals and still heavily influenced by cultural/health beliefs in their countries of origin. Or perhaps it is the case that different age groups have differentially integrated to UK society. Length of time residing in UK may be an important confounder, particularly considering the more marked inequalities observed among older age groups. Inequalities appear to be widest in those who are older in age or extremely clinically vulnerable (and as per above query - perhaps those who are more clinically vulnerable are also by default older in age because of increased risk of comorbidities?). One would also expect that the length of time individuals have been living in UK will affect/influence their acceptance of UK vaccine-related messaging, as well as likelihood of language barriers etc?

- Following on from this, did you have any information on whether some participants were asylum seekers or recently-arrived refugees? I'm sure this would only be a small minority (and likely to be very ethnically diverse, so may not allow the same level of detailed analysis), but they are a particularly important subgroup nonetheless.

- The results relating to the Bangladeshi population are interesting and warrant some further explanation or clarification. From Figure 1 C it appears that they had higher uptake of influenza vaccine vs. White British population in 2019/20. Figure 2B suggested that this was particularly the case in more deprived areas. This alone is an interesting finding. But Table S1 suggests that influenza vaccine uptake was lower in the Bangladeshi population vs. White British population in 2019/20. If I understand correctly, this is because of differences in eligibility groups between Bangladeshi and GM population (per Table S3). It would be very useful to acknowledge this finding in the manuscript as it is not entirely intuitive.

[LINK]

---

## [Decision Letter · Decision Letter 2]

24 Jan 2022

Dear Dr. Watkinson,

Thank you very much for re-submitting your manuscript "Ethnic inequalities in Covid-19 vaccine uptake and comparison to seasonal Influenza vaccine uptake in Greater Manchester: a cohort study" (PMEDICINE-D-21-04381R2) for review by PLOS Medicine.

I have discussed the paper with my colleagues and the academic editor and it was also seen again by three reviewers. I am pleased to say that provided the remaining editorial and production issues are dealt with we are planning to accept the paper for publication in the journal.

[LINK]

We look forward to receiving the revised manuscript by Jan 31 2022 11:59PM.   

Sincerely,

Callam Davidson, 

Associate Editor 

PLOS Medicine

plosmedicine.org

Requests from Editors:

Please update your title to 'Ethnic inequalities in Covid-19 vaccine uptake and comparison to seasonal Influenza vaccine uptake in Greater Manchester, UK: a cohort study. 

The Data Availability Statement (DAS) requires revision. I feel the URL included provides detailed information for those who have provided their data for research purposes but does not provide sufficient information for researchers wishing to access the data – please provide an additional URL/email address suitable for researcher enquiries.

Please consider quantifying key findings from the ethnic inequalities in influenza vaccine update in your abstract (e.g. those found on lines 301-2). 

Please trim the limitations in your abstract (lines 30-33) down to one summarising sentence. 

Lines 177 and 225: Please name and cite items located in your supplementary materials per our guidance (https://journals.plos.org/plosmedicine/s/supporting-information) e.g. (S1 Analysis Plan). 

Please remove the data sharing agreement and data availability section from the end of the main text as this information is captured in your Data Availability Statement as part of the submission form and will be published as metadata. 

Lines 203 and 420: The terms gender and sex are not interchangeable (as discussed in http://www.who.int/gender/whatisgender/en/ ); please ensure you are using the appropriate term in the main text and the relevant figures/tables in the supplementary materials.

Comments from Reviewers:

Reviewer #2: I thank the authors for addressing my comments and have nothing more to add.

Reviewer #3: Alex McConnachie, Statistical Review

I thank the authors for their consideration of my original comments, and I am happy with their responses. I have no further comments to make.

Reviewer #4: Thank you for the opportunity to review this revised manuscript. I believe that the authors have addressed my previous queries and recommend that the article should be accepted for publication.

[LINK]

---

## [Editor Report · Decision Letter 3]

27 Jan 2022

Dear Dr Watkinson, 

On behalf of my colleagues and the Academic Editor, Dr Mirjam Kretzschmar, I am pleased to inform you that we have agreed to publish your manuscript "Ethnic inequalities in Covid-19 vaccine uptake and comparison to seasonal Influenza vaccine uptake in Greater Manchester, UK: a cohort study" (PMEDICINE-D-21-04381R3) in PLOS Medicine.

When making the formatting changes, please also make the following update:

* For consistency with the rest of the manuscript and to avoid confusion, please update the bullet at line 56 to read: 'We used electronic health records for the population of Greater Manchester, England, to estimate inequalities in Covid-19 and seasonal Influenza vaccine uptake between the White British group and 16 minority ethnic groups'.

PRESS

Sincerely, 

Callam Davidson 

Associate Editor 

PLOS Medicine